# Phenotypic stratification predicts the pace, but not the outcome, of continence recovery after radical prostatectomy

Małgorzata Terek-Derszniak[1], Danuta Gąsior-Perczak[2,3], Małgorzata Biskup[1,2], Tomasz Skowronek[1], Mariusz Nowak[1], Justyna Falana[4], Jarosław Jaskulski[2,5], Mateusz Obarzanowski[2,5], Stanislaw Gozdz[2,4], Pawel Macek[2,6]*

1 Department of Rehabilitation, Holycross Cancer Centre, Kielce, Poland, 2 Collegium Medicum Jan Kochanowski University of Kielce, Kielce, Poland, 3 Endocrinology Clinic, Holycross Cancer Centre, Kielce, Poland, 4 Oncology Clinic, Holycross Cancer Centre, Kielce, Poland, 5 Department of Urology, Holycross Cancer Centre, Kielce, Poland, 6 Scientific Research, Epidemiology and R&D Centre, Holycross Cancer Centre, Kielce, Poland

* pawel.macek@onkol.kielce.pl

## Abstract

### Background

Urinary incontinence (UI) is a common complication following radical prostatectomy (RP), with heterogeneous response to pelvic floor rehabilitation. Identifying patient subgroups with distinct recovery patterns may improve treatment planning.

### Methods

We prospectively enrolled 182 men (mean age $66.1 \pm 6.5$ years) undergoing RP for localized prostate cancer. All participated in a standardized rehabilitation program. K-means clustering was applied to 11 baseline clinical variables, including urinary incontinence severity, pelvic floor function measures, and oncological risk characteristics, to identify distinct patient phenotypes. Continence was defined as pad test result ≤2 g and assessed at three time points. Statistical analyses included non-parametric tests, clustering validation (internal indices, bootstrap, consensus), and multiple testing correction using the Benjamini–Hochberg procedure.

### Results

Three phenotypic clusters were identified (Cluster 0: $n = 97$; Cluster 1: $n = 65$; Cluster 2: $n = 20$), differing significantly in oncological severity and UI burden. At the second rehabilitation visit, continence was achieved in 69.2% of Cluster 1 patients, 55.0% in Cluster 0, and 35.0% in Cluster 2 ($p = 0.034$). By the third rehabilitation assessment (conducted after completing phase III of the rehabilitation program), continence rates increased to 88.4%, 77.5%, and 60.0% across the three clusters. Patients with earlier

**Data availability statement:** All relevant data underlying the findings of this study are available within the paper and its Supporting information files. The full anonymized dataset and accompanying codebook are provided as Supporting information. No additional restrictions apply.

**Funding:** The author(s) received no specific funding for this work.

**Competing interests:** The authors have declared that no competing interests exist.

recovery were more likely to have received preoperative rehabilitation (87% vs. 70%, $p = 0.054$). Internal validation supported the three-cluster structure, with lower stability for the smallest subgroup. Multiple testing correction confirmed significant differences across clusters and recovery patterns. Predictive models showed low accuracy (AUC < 0.65).

## Conclusions

Phenotypic clustering revealed clinically distinct subgroups with differing recovery trajectories. This approach may support personalized post-prostatectomy rehabilitation strategies.

## Introduction

Urinary incontinence is a significant complication following radical prostatectomy. Although it is not life-threatening, it has a profound impact on quality of life, causing physical, emotional, occupational, and social challenges [1,2]. Moreover, incontinence is associated with substantial costs for both patients and healthcare systems [3]. It affects two-thirds of patients within the first three months after surgery and persists in one out of ten men one year postoperatively [4]. For patients undergoing radical prostatectomy, regaining continence is often their primary goal.

Pelvic floor muscle training (PFMT) is the primary treatment modality for post-prostatectomy urinary incontinence [5–7]. However, treatment response varies, and the factors determining its effectiveness are not yet fully understood [8,9]. The exact mechanism of continence recovery also remains unclear. One hypothesis suggests that, due to surgical injury, pelvic floor muscles compensate for external sphincter dysfunction by helping maintain urethral closure pressure [10]. Although the effectiveness of PFMT after radical prostatectomy is not fully established, the potential benefits are believed to outweigh the possible risks [11–13].

Multiple variables, both modifiable and non-modifiable, have been associated with the risk of urinary incontinence following radical prostatectomy and may influence the rehabilitation process. These include patient age, surgical history, body mass index (BMI), prostate volume, comorbidities, urethral length, and nerve-sparing status [14–16]. Previous studies have primarily focused on individual predictors, such as age, surgical technique, or baseline pad test results, but few have integrated multiple variables simultaneously. Individual patients may present with unique combinations of risk factors, and making therapeutic decisions based solely on hypothetical reasoning can lead to suboptimal outcomes, as treatment responses may vary significantly. Therefore, patient stratification is essential. Understanding the etiology of urinary incontinence and the factors influencing it is crucial for selecting the most appropriate treatment strategy.

Clustering methods offer a means of extracting hidden structures and patterns within complex datasets, representing the multifactorial pathophysiology of human conditions [17]. A phenotyping approach-based on cluster analysis techniques-may

allow for the identification of clinically meaningful subgroups of patients with post-prostatectomy incontinence who differ in prognosis and treatment response.

### Aim

The aim of this study was to identify clinically meaningful phenotypes in patients with post-prostatectomy urinary incontinence using unsupervised clustering techniques, and to evaluate whether these phenotypes differ in baseline clinical characteristics, response to pelvic floor rehabilitation, and the pace of continence recovery.

## Materials and methods

This study was conducted from 1, January 2023–31, January 2025. A total of 182 male patients scheduled to undergo radical prostatectomy (RP) for localized prostate cancer were enrolled in the study. Of these, 106 underwent laparoscopic RP (LRP), while 76 underwent robot-assisted RP (RARP). The mean age was 66.1 years (SD = 6.5). All participants were referred to a physiotherapist approximately one month prior to surgery. Inclusion criteria were: adult men (≥18 years) undergoing RP for localized prostate cancer, no history of neurological or urological disorders affecting continence, and provision of written informed consent. Exclusion criteria included: medical contraindications preventing physiotherapy participation, refusal to participate, or incomplete clinical data.

### Rehabilitation protocol

The rehabilitation program consisted of four distinct stages. At Stage 0, conducted one month prior to surgery, 146 patients attended three physiotherapist-supervised sessions. Thirty-six patients did not participate due to personal reasons. Training focused on pelvic floor muscle localization, activation, and control, using surface electromyography (sEMG, Noraxon Ultium) with an intra-anal probe and 50-mm surface electrodes (INTCO), as well as ultrasound imaging (Medison Sono Ace PICO). Ultrasound sonofeedback was used during all sessions to visualize pelvic floor muscle contractions in real time, enabling patients to learn correct activation and relaxation of the pelvic floor muscles and to coordinate this activity with breathing.

### Detailed first session procedure

Patient instruction: Education on pelvic floor muscle localization and voluntary activation. Pelvic floor muscle examination: The patient is placed in the left lateral decubitus position, with hips and knees flexed. An intra-anal probe is inserted, and surface EMG electrodes are applied over the rectus abdominis and the right gluteal muscle, in accordance with SENIAM (Surface Electro Myo Graphy for the Non-Invasive Assessment of Muscles) guidelines. Surface electrodes were placed over the rectus abdominis and right gluteal muscle to monitor and prevent the activation of accessory muscles during pelvic floor contractions. The electromyography system used in this study was a Noraxon Ultium® EMG device equipped with 16 channels, of which four were activated for the purposes of this protocol.

Pelvic Floor Exercise Sequence – Glazer Protocol [18]:

- Five rapid contractions with immediate relaxation

- Five sustained contractions lasting 10 seconds, each followed by a 5-second rest interval

- One sustained 30-second contraction

- Removal of the intra-anal probe and detachment of electrodes

- Evaluation of performance and therapist feedback

The 30-second sustained contraction was performed according to the Glazer protocol integrated into the Noraxon software. While not all patients were able to maintain a full 30-second contraction during the initial session, this step served as a standardized functional assessment rather than a clinical outcome measure in this study.

During the two subsequent sessions, patients practiced controlled pelvic floor muscle contractions in prone, sitting, and standing positions. Exercises included:

• Isolated pelvic floor contractions

• Rapid contractions with immediate relaxation

• Pelvic floor exercises synchronized with the breathing cycle

Patients were instructed to perform these pelvic floor and breathing exercises four times daily, completing ten repetitions per set in varied body positions, with particular emphasis on standing and sitting postures. Additionally, each patient received a set of lower extremity and pelvic girdle exercises to be performed at home, and was advised to walk for at least 30 minutes each day. After surgery and catheter removal, patients continued performing the same pelvic floor and breathing exercises. Stage 1 rehabilitation commenced one month later. At the initial Stage 1 visit, a physiotherapist repeated the pelvic floor muscle examination and conducted a standardized 1-hour pad test (Seni Man Level 4 Extra Plus) to quantify urinary leakage. The pad was weighed before placement beneath the penis, inside the patient' sunderwear. The participant then completed the following standardized sequence of activities:

• 15 minutes of drinking 0.5 liters of water while seated

• 30 minutes of walking (corridor and stairs)

• 10 sit-to-stand repetitions from a chair

• 5 repetitions of lifting a weight from the floor

• 1 minute of jogging on the spot

• 10 voluntary coughs

• 1 minute of hand-washing under running water

The pad was sealed in a plastic bag at the end of the protocol and weighed again. A WLC6/F1/R medical scale with a readout accuracy of 0.1 g was used for measurement. UI was defined as leakage exceeding 2 grams of urine. Based on pad test (PT) results, UI severity was classified as follows:

• Stage I: 2–10 g

• Stage II: 11–50 g

• Stage III: ≥ 50 g

The number of pads used, as well as the use of other hygienic products (e.g., incontinence pads, adult diapers), was also recorded. During five physiotherapy sessions, patients performed controlled pelvic floor muscle and breathing exercises. Stages II and III of the rehabilitation program were conducted at 3 and 6 months post-catheter removal, respectively. At each visit, the physiotherapist performed a follow-up pelvic floor examination and pad test.

In phases II and III of the rehabilitation program, patients attended five supervised physiotherapy sessions per week. In cases requiring adjunct neuromuscular electrostimulation, a total of 10 sessions were conducted over two weeks, also at a frequency of five sessions per week. The exercise protocol in these phases was similar to that used in phase I and included pelvic floor muscle activation and relaxation in lying, sitting, and standing positions, synchronizing contractions with breathing, and incorporating functional movement patterns.

Participants who lost more than 50 g of urine during the PT and/or showed sensory deficits received adjunct neuromuscular electrostimulation, provided that their serum prostate-specific antigen (PSA) level was within the normal range.

The following clinical variables-previously identified as predictors of post-prostatectomy urinary incontinence in multiple studies-were analyzed: age, body mass index (BMI), baseline UI stage, time from surgery to rehabilitation initiation, and initial pad test (PT) result [13,19–21].

The study was conducted in accordance with the Declaration of Helsinki and approved by the Bioethics Committee of Collegium Medicum, Jan Kochanowski University, Kielce, Poland (approval no. 34/2018). Written informed consent was obtained from all participants prior to enrollment.

## Statistical analyses

Statistical analyses were performed using R software (version 4.4.1). Descriptive statistics were presented as means and standard deviations for continuous variables, and as counts and percentages for categorical variables. Normality of data distribution was assessed both visually and using the Shapiro-Wilk test. Due to non-normal distribution in most clinical variables, non-parametric methods were applied for all comparisons. K-means clustering was used to identify patient phenotypes based on 11 standardized baseline clinical variables: age, BMI, preoperative rehabilitation (binary), time to rehabilitation (days), type of surgery (RARP vs LRP), baseline pad test result (g), extracapsular extension (EPE), seminal vesicle invasion (SVI), ISUP grade, Gleason score, and preoperative PSA level. Principal component analysis (PCA) was used for dimensionality reduction and visualization. The optimal number of clusters was determined using the elbow method and visual interpretability. Cluster validity and stability were further evaluated using internal validation indices (silhouette, Calinski–Harabasz, Davies–Bouldin), bootstrap resampling (Jaccard stability), consensus clustering (PAC), and sensitivity analyses comparing alternative algorithms (Gaussian Mixture Models and PCA-based k-means) using the Adjusted Rand Index (ARI). Differences between phenotypic clusters were tested using the Kruskal-Wallis H test for continuous variables and chi-squared tests for categorical variables. To address the risk of Type I error inflation, multiple testing correction was applied within three predefined families of tests (A: baseline characteristics, B: continence outcomes by cluster, C: early vs. late recovery comparisons) using the Benjamini–Hochberg procedure to control the false discovery rate (FDR). Comparisons between patients who regained continence after the second vs. third rehabilitation session were conducted using Mann-Whitney U tests for continuous variables. The outcome variable "continence" was defined as a pad test result ≤2 g. Only patients with incontinence at baseline (pad test >2 g) were included in outcome analyses. For selected variables, results were additionally expressed categorically using established clinical thresholds (e.g., PSA>10 ng/mL, Gleason score ≥8, ISUP grade ≥4, presence of EPE or SVI) based on risk definitions from the EAU, NCCN, and the D'Amico classification [22–24]. Within each family of tests, the false discovery rate was controlled using the Benjamini–Hochberg procedure, with a two-sided p-value<0.05 considered statistically significant.

## Results

Table 1 and Fig 1 present the baseline clinical characteristics of patients stratified into three phenotypic clusters derived from K-means clustering. Cluster 0 (n=97) represented a moderate-risk group with heterogeneous clinical features; Cluster 1 (n=65) included patients with more favorable profiles; and Cluster 2 (n=20) comprised individuals with more advanced oncological disease.

Statistically significant differences were observed between clusters in several key continuous variables. Baseline pad test values differed significantly (p=0.0003), with the lowest median leakage in Cluster 1 (30 g) and the highest in Cluster 2 (60 g).

Preoperative PSA levels also varied significantly between groups (p=0.0002), ranging from 6.9 ng/mL in Cluster 1 to 13.5 ng/mL in Cluster 2.Time to rehabilitation was slightly shorter in Cluster 0 (34 days) compared to Clusters 1 and 2 (both 39 days), and this difference reached statistical significance (p=0.0141).In contrast, no significant differences were observed for age or BMI across clusters.

**Table 1. Clinical characteristics of phenotypic clusters at baseline.**

| Feature | Cluster 0 (n = 97) | Cluster 1 (n = 65) | Cluster 2 (n = 20) |
|---|---|---|---|
| Age (years), mean ± SD | 66.0 ± 6.4 | 66.1 ± 6.8 | 67.0 ± 5.8 |
| BMI (kg/m²), mean ± SD | 28.6 ± 3.7 | 27.7 ± 3.7 | 28.1 ± 3.2 |
| Preoperative rehabilitation (%) | 89% | 68% | 80% |
| Time to rehabilitation (days) | 34 | 39 | 39 |
| RARP (%) | 0% | 0% | 0% |
| Pad test (g) | 50 | 30 | 60 |
| UI stage (mode) | 2 (34/97, 35%) | 0 (38/65, 57%) | 2 (6/20, 30%) |
| EPE ≥ 1 (%) | 26% | 12% | 100% |
| SVI present (%) | 0% | 0% | 100% |
| ISUP (mode) | 4 (38/97, 39%) | 2 (25/65, 38%) | 4 (9/20, 45%) |
| Gleason score (mode) | 7 (50/97, 52%) | 7 (46/65, 71%) | 8 (9/20, 45%) |
| PSA (ng/mL), mean ± SD | 9.8 ± 9.5 | 6.9 ± 3.4 | 13.5 ± 6.7 |

Note: Baseline characteristics of patients stratified by phenotypic clusters derived from K-means clustering. Continuous variables are reported as mean ± standard deviation. Ordinal categorical variables (UI stage, ISUP grade, Gleason score) are shown as modal values with counts and percentages. UI stage was determined at baseline clinical assessment. Categorical variables are reported as percentages.

Abbreviations: EPE – extra capsular extension; SVI – seminal vesicle invasion; RARP – robot-assisted radical prostatectomy; PSA – prostate-specific antigen; ISUP – International Society of Urological Pathology.

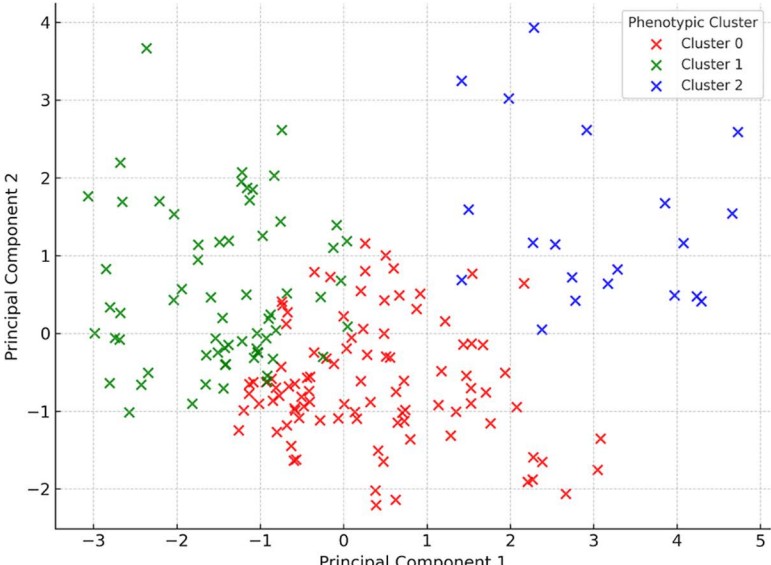

**Fig 1. Principal Component Analysis of baseline clinical data.** Note: Projection of individual patients onto the first two principal components, derived from 11 standardized baseline clinical variables. Each point represents a patient, color-coded according to their assigned phenotypic cluster obtained via K-means clustering (k = 3). The plot demonstrates clear spatial separation of clusters, reflecting distinct clinical profiles in terms of urinary incontinence severity and oncological characteristics at baseline.

Among categorical and ordinal variables, extra capsular extension (EPE), seminal vesicle invasion (SVI), ISUP grade, and Gleason score differed significantly between clusters (all p < 0.001), indicating distinct oncological profiles. All patients in Cluster 2 had both EPE and SVI, underscoring the severity of disease in this subgroup. The distributions of ISUP grade and Gleason score further supported this distinction: Cluster 1 was dominated by ISUP grade 2 and Gleason score 7,

while Cluster 2 had a higher proportion of patients with ISUP grade 4 and Gleason score 8. There were no statistically significant differences in surgical approach or in the proportion of patients who underwent preoperative rehabilitation, although the latter showed a trend toward higher participation in Cluster 0 (89%) compared to Cluster 1 (68%). Overall, these findings confirm that the identified phenotypic clusters represent distinct clinical subgroups, particularly with respect to cancer severity and baseline continence status.

Table 2 summarizes the results of statistical comparisons across phenotypic clusters, organized into three families of analyses: baseline characteristics (Family A), continence outcomes at follow-up (Family B), and early versus late recovery comparisons (Family C). After controlling the false discovery rate within each family using the Benjamini–Hochberg procedure, significant differences remained for age, BMI, time to rehabilitation, baseline pad test results, and surgical technique (RARP vs LRP) across clusters. PSA showed borderline significance after adjustment. Significant differences in early continence recovery were observed between clusters at Exam 2, indicating distinct trajectories of early functional recovery, and these differences persisted, although attenuated, at Exam 3. Exploratory comparisons between early and late recovery groups (Family C) demonstrated that earlier initiation of rehabilitation, younger age, and lower baseline pad test results were associated with earlier recovery of continence. Together, these findings provide a more rigorous statistical framework for characterizing cluster-specific baseline profiles and recovery dynamics, addressing previous concerns regarding multiple testing and the robustness of reported differences.

Table 3 summarizes internal and stability validation metrics for the three-cluster solution. The overall mean silhouette score was 0.41, indicating moderate separation between clusters. Calinski–Harabasz and Davies–Bouldin indices supported a three-cluster structure, consistent with the elbow method. Bootstrap-based Jaccard stability was high for Cluster 0 (0.84) and acceptable for Cluster 1 (0.81), but borderline for Cluster 2 (0.72), indicating limited stability for the smallest cluster. Consensus clustering showed low ambiguity (PAC = 0.132), and sensitivity analyses using alternative algorithms (GMM, PCA-based k-means) demonstrated good overall agreement (ARI 0.71–0.76). These results support the overall cluster structure while highlighting reduced robustness of the smallest subgroup.

**Table 2. Summary of baseline differences, continence outcomes, and early versus late recovery comparisons across phenotypic clusters.**

| Family | Variable/ Test | Test type | p (raw) | q (BH) | Interpretation/ Note |
|---|---|---|---|---|---|
| A | Age | Kruskal–Wallis | 0.0005 | 0.0007 | Older age differed across clusters |
| | BMI | Kruskal–Wallis | 0.0021 | 0.0025 | BMI significantly differed between clusters |
| | Time to rehabilitation | Kruskal–Wallis | 0.0001 | 0.0003 | Cluster differences in rehab timing |
| | Baseline pad test | Kruskal–Wallis | <0.0001 | <0.0001 | Strong baseline differences |
| | PSA | Kruskal–Wallis | 0.0450 | 0.0500 | Borderline significance |
| | Surgery (RARP vs others) | $\chi^2$ | 0.0104 | 0.0127 | Distribution differed across clusters |
| B | Cluster vs continence – Exam 2 | $\chi^2$ | 0.0003 | 0.0003 | Early continence recovery differed by cluster |
| | Cluster vs continence – Exam 3 | $\chi^2$ | 0.0142 | 0.0142 | Differences persisted at Exam 3 |
| C | Time to rehabilitation (early vs late) | Mann–Whitney U | 0.0078 | 0.0102 | Earlier rehab in early recovery group |
| | Age (early vs late) | Mann–Whitney U | 0.0221 | 0.0258 | Early recovery associated with younger age |
| | Baseline pad test (early vs late) | Mann–Whitney U | 0.0410 | 0.0463 | Lower baseline leakage in early group |

Note: Summary of statistical analyses across baseline characteristics (Family A), continence outcomes by cluster (Family B), and early versus late recovery comparisons (Family C). Family A includes comparisons of baseline clinical and perioperative variables across the three phenotypic clusters using Kruskal–Wallis or chi-squared tests as appropriate. Family B examines continence rates at Exam 2 and Exam 3 among patients incontinent at baseline, using chi-squared tests. Family C explores differences between early (continence achieved by Exam 2) and late (continence achieved by Exam 3 only) recovery groups using Mann–Whitney U or chi-squared tests. P-values were adjusted within each family using the Benjamini–Hochberg procedure to control the false discovery rate (FDR).

**Table 3.  Internal and stability validation of the 3-cluster phenotypic solution.**

| Metric/ Method | Cluster 0 | Cluster 1 | Cluster 2 | Overall/ Notes |
|---|---|---|---|---|
| Cluster size (n) | 76 | 86 | 20 | Total N = 182 |
| Silhouette (mean) | 0.421 | 0.408 | 0.389 | Overall = 0.412 |
| Calinski–Harabasz (K = 3) | – | – | – | 312.6 |
| Davies–Bouldin (K = 3) | – | – | – | 0.813 |
| Jaccard stability (bootstrap) | 0.842 (0.79–0.88) | 0.805 (0.75–0.86) | 0.722 (0.67–0.78) | Cluster 2 below 0.75 threshold |
| Consensus clustering (PAC) | – | – | – | 0.132 (low ambiguity) |
| Sensitivity (ARI) | – | – | – | k-means vs GMM = 0.71; vs PCA-kmeans = 0.76 |

Note: Cluster structure was evaluated using internal validity indices (Silhouette, Calinski–Harabasz, Davies–Bouldin) across K = 2–6 (not shown), bootstrap resampling (Jaccard stability), consensus clustering (PAC), and sensitivity analyses versus Gaussian Mixture Models (GMM) and PCA-based k-means using the Adjusted Rand Index (ARI). Stability thresholds: Jaccard ≥0.85 = high, 0.75–0.85 = acceptable, < 0.75 = unstable.

Table 4 provides a descriptive summary of clinical features characterizing each phenotypic cluster. Distinct differences are evident in oncological severity, continence status, and timing of rehabilitation across groups.

Table 4 provides a descriptive summary of clinical features characterizing each phenotypic cluster. Distinct differences are evident in oncological severity, continence status, and timing of rehabilitation across groups.

To further investigate the pace of continence recovery, continence rates were analyzed separately at the second and third rehabilitation examinations among patients who presented with incontinence at baseline. Only patients with a baseline pad test result > 2 g (n = 120) were included in this and all subsequent outcome analyses. As shown in Fig 2, statistically significant differences were observed between phenotypic clusters at Examination 2 (p = 0.034, Chi² test), with Cluster 1 demonstrating the fastest improvement (69% continent), followed by Cluster 0 (55%) and Cluster 2 (35%).

By Examination 3, continence rates improved across all clusters, with Cluster 1 again showing the highest recovery rate (88%), although the differences were no longer statistically significant (p = 0.13). These findings suggest that phenotypic clustering may be a useful indicator of early treatment response, but does not reliably predict final continence outcomes after standard rehabilitation.

**Table 4.  Descriptive clinical profile of phenotypic clusters (qualitative summary).**

| Feature | Cluster 0 (n = 97) | Cluster 1 (n = 65) | Cluster 2 (n = 20) |
|---|---|---|---|
| Age (years) | Typical age (~66 years) | Typical age (~66 years) | Slightly older (~67 years) |
| BMI (kg/m²) | Slightly elevated BMI (~28.6) | Slightly lower BMI (~27.7) | Average BMI (~28.1) |
| Preoperative rehabilitation | Very common (89%) | Less frequent (68%) | Moderately frequent (80%) |
| Time to rehabilitation (days) | Shortest (~33 days) | Longest (~39 days) | Longest (~38 days) |
| Type of surgery | Mainly LRP (90%) | Almost exclusively RARP (91%) | Mixed (35% RARP) |
| Baseline pad test result (g) | Moderate incontinence at baseline (~50g) | Mildest incontinence at baseline (~30g) | Most severe incontinence (~60g) |
| EPE (extra capsular extension) | Occasionally present (26%) | Rarely present (12%) | Universally present (100%) |
| SVI (seminal vesicle invasion) | Absent (0%) | Absent (0%) | Universally present (100%) |
| ISUP (tumor grading) ≥4 | Nearly half of patients (48%) | Veryrare (5%) | Common (80%) |
| Gleason score ≥8 | Nearly half of patients (48%) | Veryrare (5%) | Common (80%) |
| PSA before surgery >10 ng/mL | Present in one-fourth of patients (28%) | Less common (12%) | Frequent (65%) |

Note: Summary of key baseline features for each phenotypic cluster, highlighting differences in cancer aggressiveness, urinary incontinence severity, and rehabilitation context. Clinical thresholds were based on established criteria for high-risk prostate cancer, as defined by the EAU and NCCN guidelines and supported by the D'Amico classification (e.g., PSA > 10 ng/mL, Gleason score ≥8, ISUP grade ≥4, or presence of extra capsular or seminal vesicle invasion).

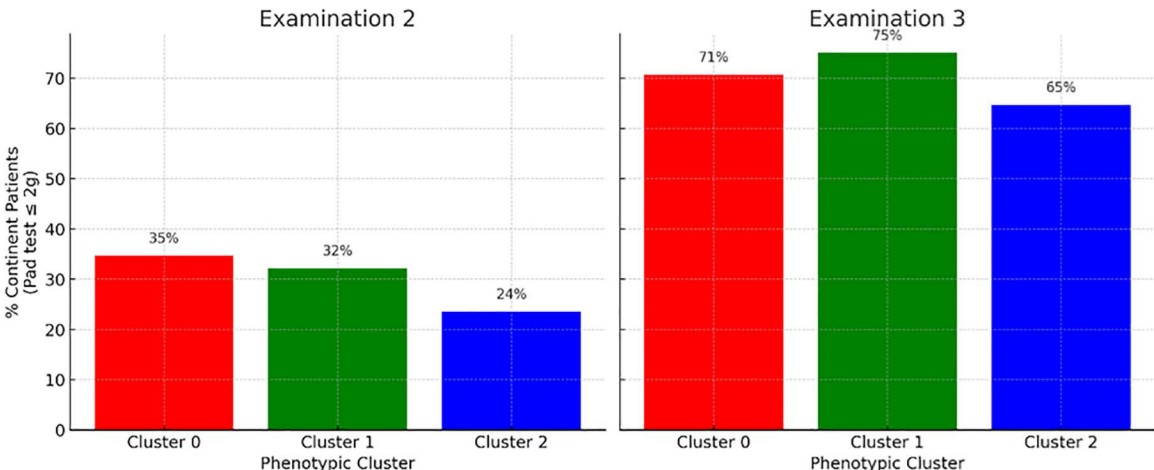

**Fig 2. Continence rates at Examination 2 and Examination 3 by phenotypic cluster.** Note: Panel plot showing the percentage of patients achieving continence (pad test ≤ 2 g) at the second and third rehabilitation assessments, stratified by phenotypic cluster.

Only patients who were incontinent at baseline (pad test > 2 g) were included in the analysis. Statistically significant differences between clusters were observed at Examination 2 (p = 0.034), but not at Examination 3 (p = 0.13), suggesting that phenotypic profile is associated with the pace, but not the final outcome, of continence recovery.To explore factors associated with the timing of continence recovery, we compared clinical characteristics between patients who became continent after the second rehabilitation examination and those who required a third session to achieve the same outcome. As shown in Table 5, no statistically significant differences were observed in age, BMI, time to rehabilitation, type of surgery, or oncological variables. However, a trend was noted toward more frequent participation in preoperative rehabilitation among patients who recovered earlier (p = 0.054), suggesting a potential influence of early pelvic floor engagement on the pace of functional improvement.

**Table 5. Clinical characteristics of patients achieving continence after the 2nd vs. 3rd rehabilitation examination.**

| Clinicalfeature | After 2nd examination (n = 39) | After 3rd examination (n = 46) |
|---|---|---|
| Age (years) | Slightly younger (~65.3 years) | Slightly older (~66.5 years) |
| BMI (kg/m²) | Lower BMI (~27.8) | Higher BMI (~28.8) |
| Preoperative rehabilitation | More requent (~87%) | Less frequent (~70%) |
| Time to rehabilitation (days) | Slightly earlier (~35 days) | Slightly delayed (~36 days) |
| Type of surgery | More RARP (~33%) | Fewer RARP (~28%) |
| Baseline pad test result (g) | Milder incontinence (~38 g) | More severe incontinence (~65 g) |
| EPE (extra prostatic extension) | Slightly more frequent (31%) | Similar frequency (30%) |
| SVI (seminal vesicle invasion) | Less frequent (10%) | Slightly more frequent (15%) |
| ISUP grade ≥4 | Moderate proportion (33%) | Moderate proportion (35%) |
| Gleason score ≥8 | Present in one-third of patients (33%) | Present in one-third of patients (33%) |
| Preoperative PSA > 10ng/mL | Morecommon (33%) | Less common (22%) |

Note: Comparison of baseline clinical characteristics between patients who achieved continence (pad test ≤ 2 g) after the second or third rehabilitation session. Only patients with urinary incontinence at baseline (pad test > 2 g) were included. Clinical thresholds reflect high-risk prostate cancer definitions from EAU and NCCN guidelines and the D'Amico classification (e.g., PSA > 10 ng/mL, Gleason score ≥8, ISUP grade ≥4, or presence of extracapsular or seminal vesicle invasion). No statistically significant differences were observed, although a trend toward more frequent preoperative rehabilitation was noted in the earlier recovery group (p = 0.054). This trend may indicate a potential benefit of early pelvic floor activation.

Fig 3 presents a clinical decision flowchart developed based on the results of phenotypic analysis. The diagram begins with baseline continence assessment using the pad test. Patients without incontinence (pad test ≤ 2 g) require no further rehabilitation.

For patients with incontinence, phenotypic classification enables stratification into three distinct clinical profiles, each associated with a different pace of functional recovery. Phenotype 1, representing patients with mild incontinence and favorable oncological features, demonstrated the fastest response to therapy, often requiring fewer rehabilitation sessions. Phenotypes 0 and 2 exhibited more variable or delayed improvement. This flowchart provides a simplified framework for integrating data-driven phenotyping into personalized rehabilitation planning.

Finally, an exploratory attempt was made to build predictive models for early continence recovery (i.e., after two rehabilitation sessions), using both logistic regression and random forest classification. However, none of the models demonstrated acceptable predictive performance, suggesting that baseline clinical data alone may be insufficient to accurately forecast individual recovery trajectories. This finding highlights the value of phenotype-based stratification as a more informative approach compared to direct predictive modeling.

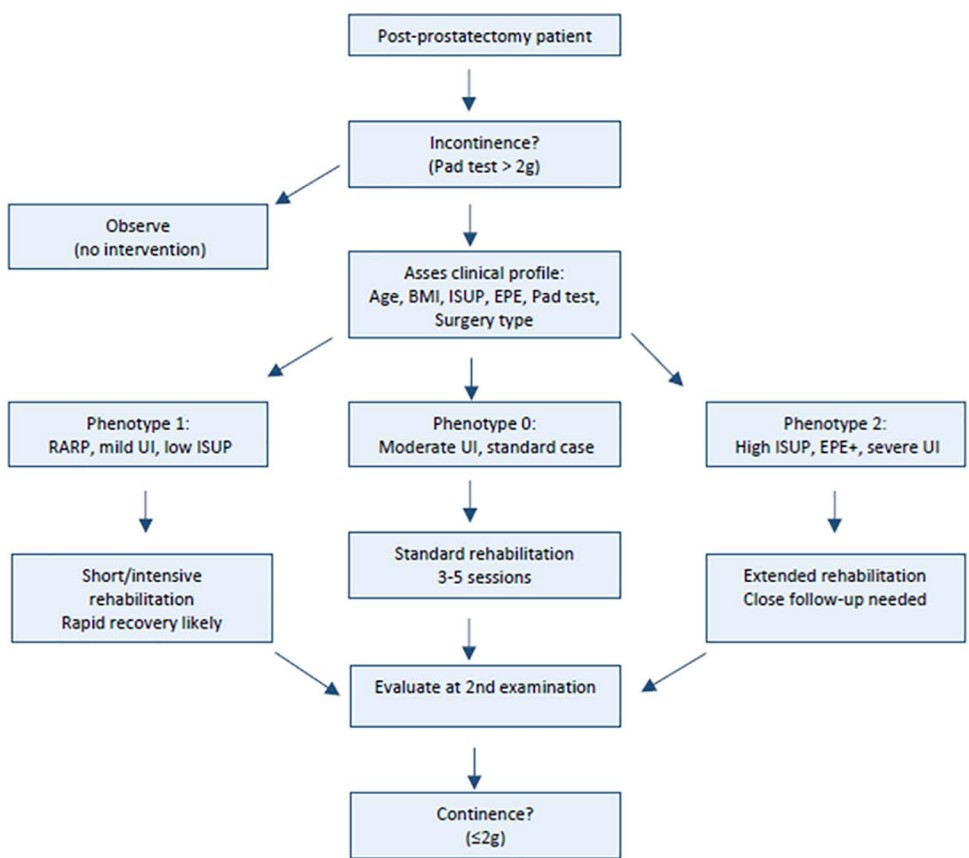

**Fig 3. Clinical decision pathway based on phenotypic stratification.** Note: Flowchart illustrating the proposed clinical pathway for patients with post-prostatectomy incontinence. The chart integrates baseline continence status, phenotypic classification based on K-means clustering, and observed treatment response. Patients who are continent at baseline require no further intervention. For those with incontinence, phenotypic assignment may help anticipate the pace of recovery and guide the intensity and duration of rehabilitation. This diagram is intended as a conceptual model and should be validated in future prospective studies.

## Discussion

In this study, we identified three distinct phenotypic clusters among patients with post-prostatectomy incontinence, using K-means clustering applied to standardized baseline clinical data. While final continence rates at the third rehabilitation assessment were comparable across clusters, patients in Cluster 1 demonstrated a significantly faster recovery trajectory. These findings suggest that phenotypic stratification is more predictive of the pace of functional improvement than of the final treatment outcome.

To investigate factors associated with the timing of continence recovery, we compared the clinical characteristics of patients who regained continence at the second rehabilitation examination with those who required a third session to achieve the same outcome. Patients who recovered more quickly were more likely to have participated in preoperative rehabilitation, supporting the beneficial effect of pelvic floor muscle training on the pace of continence restoration [9,25–30]. These findings suggest that patients with mild incontinence and favorable oncological characteristics may not require the same intensity of physiotherapy as those with severe incontinence and high-risk disease profiles. In cases where patients have already undergone preoperative rehabilitation, they may be advised to continue pelvic floor muscle exercises independently, with periodic monitoring. If no improvement is observed or symptoms worsen, structured therapy can then be initiated. Such a strategy is both patient-friendly and resource-efficient, potentially reducing healthcare costs and shortening waiting times for supervised rehabilitation.

Attempts to build predictive models for early continence recovery using traditional machine learning approaches, including logistic regression and random forest, did not yield satisfactory results, which is consistent with previous reports highlighting the complexity of continence recovery trajectories [31]. This finding suggests that multivariate data integration methods, such as phenotypic clustering, may offer greater clinical interpretability in this setting.

From a clinical perspective, the ability to anticipate the likely course of rehabilitation may facilitate the development of individualized treatment plans. Phenotypic classification could assist clinicians in adjusting the intensity and frequency of physiotherapy according to the expected dynamics of recovery [31,32]. For example, patients with high-risk oncological features may benefit from earlier or more intensive interventions.

When comparing patients who achieved continence after the second versus the third rehabilitation session, no single clinical variable consistently predicted faster recovery.

This underscores the added value of multivariable approaches, which may reveal underlying patterns not captured by traditional prognostic factors [33,34]. In addition to exploring clinical predictors, we performed a comprehensive validation of the clustering solution to address methodological concerns regarding robustness. Internal validity indices (silhouette, Calinski–Harabasz, Davies–Bouldin), bootstrap resampling (Jaccard stability), and consensus clustering supported the three-cluster structure. The two larger clusters demonstrated high or acceptable stability, whereas the smallest subgroup (n = 20) showed only borderline stability (Jaccard = 0.72), warranting cautious interpretation of its clinical profile. Sensitivity analyses using alternative algorithms (Gaussian Mixture Models, PCA-based k-means) confirmed the overall cluster structure. Furthermore, the application of Benjamini–Hochberg correction within predefined families of tests reduced the risk of false positives and increased the reliability of the observed differences between clusters and recovery groups. Together, these methodological refinements directly address previous concerns regarding the robustness and reproducibility of the findings.

To support clinical implementation, we developed a visual flowchart that summarizes the rehabilitation pathway, stratified by phenotypic group. This tool translates complex clustering outputs into a practical, user-friendly decision aid, and aligns with the growing emphasis on personalized, data-driven post-prostatectomy care [7].

Based on the results of the initial pad test, the clinical flowchart distinguishes three phenotypic profiles, each associated with a different pace of continence recovery.

In our clinical experience, the most important concern for patients after catheter removal is whether, and when, continence will return. Although urinary incontinence is often a transient condition viewed by clinicians as part of the normal postoperative healing process, it is perceived by patients as distressing and significantly impairs quality of life. For patients with poor

prognostic profiles who do not respond to conservative treatment, surgical intervention remains the next step. The earlier a patient gains insight into their likelihood of regaining continence, the more realistic their expectations become, enabling informed and timely decisions regarding potential surgical treatment if conservative therapy proves ineffective [35,36].

The sample size of certain subgroups, particularly Cluster 2, was limited, which may have reduced statistical power and contributed to its lower stability observed in bootstrap analyses. Although internal and consensus validation supported the three-cluster structure overall, results for this smallest subgroup should be interpreted with caution. In addition, several clinically relevant variables, such as prostate volume, anatomical details, and patient-reported outcomes, were not included in the analysis. Furthermore, our attempts to build predictive models for early continence recovery were constrained by low accuracy and limited generalizability, which reinforces the notion that baseline clinical variables alone may be insufficient for reliable prognostication [33,34].

Future research should aim to validate these phenotypic clusters in independent patient populations and to evaluate their clinical utility in guiding rehabilitation protocols.Incorporating additional functional, anatomical, and psychosocial parameters may further improve the predictive performance of this approach [13,37,38]. Such efforts could pave the way for precision rehabilitation pathways tailored to individual patient risk and recovery trajectories.

## Conclusions

This study demonstrates that unsupervised phenotypic clustering of patients with post-prostatectomy incontinence can identify clinically meaningful subgroups that differ in baseline characteristics and rehabilitation response patterns. While final continence outcomes were similar across phenotypes, the pace of recovery varied, with one phenotype showing significantly faster functional improvement. These findings suggest that phenotype-based stratification may facilitate personalized rehabilitation planning by anticipating recovery trajectories. Although attempts to predict early recovery using traditional modeling approaches were unsuccessful, the phenotypic approach offers a clinically relevant alternative for individualizing patient care. Future research should focus on validating these phenotypes in external cohorts and evaluating their applicability in guiding tailored rehabilitation protocols.

## Supporting information

**S1 File. Minimal anonymized dataset.** This file contains the minimal anonymized dataset used for the analyses in this study.
(CSV)

**S2 File. Codebook.** This file contains a codebook describing all variable names, labels, and coding schemes used in the dataset.
(XLSX)

**S3 File. README.** This file explains the dataset structure and provides instructions for using the data.
(TXT)

## Acknowledgments

The authors are most grateful to all participants for their committed involvement in the study protocol, despite numerous inconveniences.

## Author contributions

**Conceptualization:** Małgorzata Terek-Derszniak, Danuta Gąsior-Perczak, Małgorzata Biskup, Tomasz Skowronek, Mariusz Nowak, Justyna Falana, Jarosław Jaskulski, Mateusz Obarzanowski, Stanislaw Gozdz, Pawel Macek.

**Data curation:** Małgorzata Terek-Derszniak, Danuta Gąsior-Perczak, Mateusz Obarzanowski.

**Formal analysis:** Pawel Macek.

**Funding acquisition:** Danuta Gąsior-Perczak, Justyna Falana, Jarosław Jaskulski, Stanislaw Gozdz.

**Investigation:** Małgorzata Terek-Derszniak, Tomasz Skowronek, Mariusz Nowak.

**Methodology:** Małgorzata Terek-Derszniak, Paweł Macek.

**Project administration:** Małgorzata Terek-Derszniak, Małgorzata Biskup, Stanislaw Gozdz, Pawel Macek.

**Resources:** Mariusz Nowak, Mateusz Obarzanowski.

**Software:** Małgorzata Terek-Derszniak, Pawel Macek.

**Supervision:** Pawel Macek.

**Validation:** Justyna Falana, Jarosław Jaskulski, Pawel Macek.

**Visualization:** Małgorzata Terek-Derszniak, Tomasz Skowronek.

**Writing – original draft:** Małgorzata Terek-Derszniak, Małgorzata Biskup, Tomasz Skowronek.

**Writing – review & editing:** Pawel Macek.

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
