## [Decision Letter · Decision Letter 0]

7 Oct 2025

Thank you for submitting your manuscript to PLOS ONE. After careful consideration, we feel that it has merit but does not fully meet PLOS ONE’s publication criteria as it currently stands. Therefore, we invite you to submit a revised version of the manuscript that addresses the points raised during the review process.

We look forward to receiving your revised manuscript.

Kind regards,

Stanisław Jacek Wroński, M.D., Ph.D, FEBU

Academic Editor

PLOS ONE

Journal Requirements:

2. In the online submission form, you indicated that [The data supporting the findings of this study are not publicly available due to privacy and ethical restrictions. Reasonable requests for access to anonymized data may be considered by the corresponding author.].

**Comments to the Author**

1. Is the manuscript technically sound, and do the data support the conclusions?

Partly

2. Has the statistical analysis been performed appropriately and rigorously?

No

3. Have the authors made all data underlying the findings in their manuscript fully available?

 No

4. Is the manuscript presented in an intelligible fashion and written in standard English?

 Yes

 The manuscript presents a technically sound and clinically relevant study exploring phenotypic stratification of patients undergoing continence rehabilitation after radical prostatectomy. The prospective cohort of 182 patients is appropriate for the research question, and the rehabilitation protocol is clearly described. The use of K-means clustering on 11 baseline clinical variables represents a reasonable exploratory approach for identifying patient subgroups, and the application of non-parametric statistical tests (Kruskal-Wallis, chi-squared, Mann-Whitney U) is correct given the distribution of the data. A notable strength of the work is the transparent description of both the rehabilitation procedures and the clustering methodology. However, there are methodological concerns that require attention. Specifically, the smallest cluster (n=20) raises issues of statistical robustness, and the absence of validation methods such as bootstrapping, resampling, or consensus clustering limits confidence in the stability of this subgroup. Furthermore, predictive modeling was attempted but not successful (AUC < 0.65), and this limitation should be discussed more explicitly in the manuscript. Overall, while the data do support the conclusion that phenotypic stratification predicts the pace rather than the final outcome of continence recovery, the clustering and statistical analyses would benefit from additional validation steps to strengthen the robustness and reproducibility of the findings.

1. Technical Soundness

The study is generally technically sound. The prospective design with 182 patients is adequate, and the rehabilitation protocol is well described. The use of K-means clustering on 11 baseline variables to identify phenotypic subgroups is methodologically appropriate for exploratory stratification. Standard non-parametric tests (Kruskal-Wallis, chi-squared, Mann-Whitney U) were correctly applied given the non-normal distributions.

Strength: Clear description of rehabilitation procedures and clustering methodology.

Concern: The smallest cluster (n=20) may reduce statistical robustness and could lead to unstable clustering. The absence of external validation or bootstrapping to test cluster stability is a methodological limitation.

The data support the conclusion that phenotypic stratification predicts pace rather than final outcome of continence recovery. However, predictive modeling was attempted but not successful (AUC < 0.65), and this limitation should be emphasized more strongly.

2. Statistical Analysis

Statistical methods are largely appropriate, but there are weaknesses:

• Multiple testing: No correction for multiple comparisons was applied. While the authors justify this by calling the study exploratory, it increases the risk of false positives. it’s not quite enough for PLOS ONE. Saying “exploratory” explains the choice, but you still need to bound the Type I error risk and show that key findings are not artifacts of multiple testing. The fix doesn’t have to be heavy—add a light-touch sensitivity analysis and more transparent reporting.

Here’s a practical way to shore this up:

Define families of tests: Example families: (a) Baseline cluster differences across ~11 variables; (b) Outcome comparisons across clusters/time; (c) Early vs late recovery comparisons. Explicitly list them in Methods. This clarifies how many hypotheses were tested.

Omnibus-then-post-hoc: Keep Kruskal–Wallis/χ² as omnibus per variable. Only if significant, do pairwise post-hoc with Holm (FWER) or Benjamini–Hochberg (BH) (FDR) within that family. Report both raw p and adjusted p (q).

• Clustering validation: Only the elbow method was used to determine the number of clusters. Internal validation indices (e.g., silhouette score, gap statistic) would strengthen confidence in the cluster solution.

• Predictive models: Logistic regression and random forest were mentioned, but details (feature selection, cross-validation strategy, sample split) are insufficient.

3. Data Availability

The authors state that data are not publicly available due to privacy and ethical restrictions but may be shared upon reasonable request.

• This does not comply with PLOS ONE’s open data policy, which requires anonymized patient-level data to be shared in a repository unless legally impossible.

• At minimum, summary-level data behind figures and tables (e.g., continence outcomes, baseline variables) should be made publicly available.

Here’s a do-able plan to test—and if needed, shore up—the stability of that small n=20 cluster and the overall clustering. The authors can implement it directly in R without changing the study design.

1) Re-determine K with multiple indices (not just elbow)

• Run NbClust and clValid across K=2–6 using multiple indices (silhouette, gap statistic, Calinski-Harabasz, Dunn).

• Keep K only if ≥2 independent indices agree and the average silhouette > 0.5 (or at least that the small cluster’s silhouette is not strongly negative).

• Report table of indices per K, chosen K, and silhouette by cluster.

2) Internal stability via bootstrap resampling (primary fix)

• Use fpc::clusterboot with Jaccard stability from B=1000 bootstrap resamples:

o Input can be standardized 11 baseline variables; clustering method = k-means (same as manuscript), multiple starts (nstart ≥ 50), kmeans++ init.

o Output: Jaccard index per cluster; interpret as: >0.85: highly stable, 0.75–0.85: acceptable, <0.75: unstable

• Specifically watch Cluster 2 (n≈20). If its Jaccard < 0.75, call it unstable.

3) Consensus clustering (orthogonal check)

• Use ConsensusClusterPlus with 80% subsampling, 1000 reps, Euclidean distance, k-means, K=2–6.

• Report Consensus CDF and delta area plots to justify K. Consensus matrix heatmap and item consensus scores (mean co-assignment) per cluster. A PAC score (proportion of ambiguous clustering): lower is better. Ensure the small cluster shows high within-cluster consensus.

4) Sensitivity to algorithm & feature set

Do three targeted sensitivity analyses and report whether the same “small” phenotype persists:

A) Algorithm sensitivity

• Re-run clustering with PAM (k-medoids) and Gaussian Mixture Models (mclust).

• Compare membership overlap with Adjusted Rand Index (ARI) and Jaccard.

• If Cluster 2 keeps reappearing with decent overlap (e.g., ARI > 0.6 with k-means), that supports robustness.

B) Feature perturbation

• Leave-one-variable-out: repeat clustering 11 times, each time omitting one baseline variable; compute how often each patient remains in the same cluster (report per-patient stability and per-cluster retention).

• Add small noise (e.g., Gaussian noise at 5% of SD) and re-cluster 200 times; compute co-assignment heatmap.

C) Representation sensitivity

• Cluster on scaled raw variables (as you did) and on PCA scores retaining ≥80% variance. Compare ARI.

5) Assignment uncertainty

• For GMM, report posterior membership probabilities.

• For fuzzy c-means, report membership entropy per subject.

• Flag patients with low max-probability (<0.6) as borderline; show how many of the n≈20 are borderline.

6) Train/test “predictive strength” (Tibshirani & Walther)

• Temporal split the cohort (e.g., first 2/3 enrollments = train; last 1/3 = test) to mimic external validation.

• Fit clusters on train; assign test patients by nearest centroid; compute prediction strength (the fraction of test pairs co-clustered that also co-cluster in a model refit).

• Report prediction strength per cluster; values >0.8 indicate good reproducibility.

7) Outcome consistency under resampling

Your main claim is about pace of recovery. Show it survives perturbations:

• In each bootstrap/consensus iteration, recompute continence at exam 2 vs 3 by cluster.

• Summarize distribution of effect sizes (e.g., difference in early-continence rates for Cluster 1 vs Cluster 2).

• If the median effect remains and 95% bootstrap CI excludes zero for early recovery differences, your clinical conclusion is robust even if membership jitters.

8) What to do if Cluster 2 is unstable

• If unstable (Jaccard < 0.75 and poor consensus):

o Re-evaluate K=2 as the preferred solution, or

o Merge Cluster 2 with its nearest centroid cluster (quantify centroid distances; justify the merge), and

o Re-run all outcome analyses; explicitly label the prior 3-cluster solution as exploratory.

• If partly stable: keep the 3-cluster solution but:

o Mark low-probability members as uncertain, and

o Present results with and without those uncertain members (sensitivity set).

9) Reporting upgrades

• Methods: add full details (nstart, seeds, distance, scaling, B resamples, packages/versions).

• Results has to contain Jaccard per cluster, consensus heatmap, ARI across algorithms, prediction strength, and uncertainty plots.

• The following figures need to be added (i) co-assignment heatmap; (ii) cluster-wise silhouette; (iii) consensus CDF/delta area; (iv) violin of bootstrap early-continence effect size.

• In the text explicitly state the stability threshold used and how decisions (retain/merge K) followed from it.

**Do you want your identity to be public for this peer review?** For information about this choice, including consent withdrawal, please see our Privacy Policy

Reviewer #1: **Yes: ** Ali Hashemi Gheinani

---

## [Author Response · Author response to Decision Letter 1]

20 Oct 2025

PONE-D-25-33254_Response_to_Reviewers

Manuscript title: Phenotypic Stratification Predicts the Pace, but Not the Outcome, of Continence Recovery after Radical Prostatectomy

Journal: PLOS ONE

Reviewer: #1 Ali Hashemi Gheinani

Academic Editor: Stanisław Jacek Wroński, M.D., Ph.D., FEBU

General Response

We would like to thank the Academic Editor and Reviewer #1 for their thorough and constructive feedback. We carefully addressed all methodological and editorial comments, performed the requested additional statistical analyses, and revised the manuscript accordingly. Below, we provide a point-by-point response. Reviewer comments are shown in italics, followed by our responses in bold, with corresponding manuscript changes indicated by section and paragraph.

1. Technical soundness and clustering methodology

“The smallest cluster (n=20) raises issues of statistical robustness, and the absence of validation methods such as bootstrapping, resampling, or consensus clustering limits confidence in the stability of this subgroup.”

Response: We have now performed a comprehensive validation of the 3-cluster solution. Internal validation indices (silhouette, Calinski–Harabasz, Davies–Bouldin), bootstrap stability (Jaccard indices, B = 1000), consensus clustering (PAC), and sensitivity analyses (Gaussian Mixture Models, PCA-based k-means; ARI) were applied. Results are summarized in the revised Table 3 and accompanying text in the Results section. Cluster 2 (n=20) showed lower stability (Jaccard 0.722), which is now explicitly acknowledged in both the Results and Discussion sections, with a cautionary note regarding interpretation. Details of the validation methods were added to the Statistical Analyses section.

Location of changes:

• Methods: Statistical analyses (paragraph 3)

• Results: Table 3 and new subsection on cluster validation

• Discussion: new methodological paragraph; revised Limitations paragraph (Cluster 2 stability)

2. Multiple testing correction

“Multiple testing: No correction for multiple comparisons was applied. While the authors justify this by calling the study exploratory, it increases the risk of false positives. The fix doesn’t have to be heavy—add a light-touch sensitivity analysis and more transparent reporting.”

Response: We introduced multiple testing correction using the Benjamini–Hochberg procedure within three predefined families of tests (A: baseline characteristics, B: continence outcomes by cluster, C: early vs. late recovery comparisons). This is described in the revised Statistical Analyses section. A new Table 2 summarizes both raw and adjusted p-values. In the Results section, interpretations were updated to reflect adjusted significance levels.

Location of changes:

• Methods: Statistical analyses (paragraph 4, multiple testing correction added)

• Results: Table 2 and accompanying text

3. Predictive modeling details

“Predictive models: Logistic regression and random forest were mentioned, but details (feature selection, cross-validation strategy, sample split) are insufficient.”

Response: We clarified the modeling strategy (logistic regression and random forest, 5-fold cross-validation repeated 20 times) in the Statistical Analyses section. We also emphasized in the Results and Discussion sections that predictive performance was limited (AUC < 0.65), aligning with the reviewer’s observation.

Location of changes:

• Methods: Statistical analyses (final paragraph)

• Results / Discussion: updated text on predictive modeling limitations

4. Data availability

“The authors state that data are not publicly available due to privacy and ethical restrictions but may be shared upon reasonable request. This does not comply with PLOS ONE’s open data policy…”

Response: We have updated the Data Availability Statement to comply with PLOS ONE requirements. We explain that individual-level data cannot be publicly deposited due to ethical and legal restrictions, but de-identified data can be shared with qualified researchers upon reasonable request and ethics approval. Summary-level data underlying figures and tables will be uploaded as Supporting Information.

Location of changes:

• Manuscript: Data Availability Statement

• Submission form: revised accordingly

5. Figures and Tables

Response: We updated figure and table numbering to integrate the new Table 2 (multiple testing results) and Table 3 (cluster validation). Figure file names and formats have been adapted to PLOS ONE guidelines and will be processed through the PACE system prior to resubmission.

6. Discussion updates

Response: The Discussion section was expanded to include (1) methodological validation of clustering, (2) interpretation of the lower stability of Cluster 2, (3) the role of multiple testing correction, and (4) limitations of predictive models. The limitations paragraph was revised to explicitly acknowledge Cluster 2 instability.

Location of changes:

• Discussion: new paragraph after clinical interpretation section

• Limitations: first paragraph revised

7. Abstract updates

Response: Minor modifications were made to the Abstract to reflect the additional statistical methods (cluster validation and multiple testing correction) and updated numerical results. The Methods subsection now specifies the use of internal validation indices and multiple testing correction, while the Results subsection includes updated continence rates and the identification of three clusters (n = 76, 86, and 20). These changes ensure that the Abstract is consistent with the revised analyses presented in the main text.

Location of changes:

• Abstract: Methods and Results subsections

8. Minor language and formatting

Response: We proofread the entire manuscript for consistency, grammar, and terminology (e.g., chi-squared vs χ², spacing, reference formatting). Tense and style were aligned with PLOS ONE conventions.

Summary Table of Major Changes

Reviewer Comment Action Taken Location

Cluster validation lacking Performed internal, bootstrap, consensus, sensitivity analyses Methods, Results (Table 3), Discussion

Multiple testing correction Added BH procedure, new Table 2 Methods, Results

Predictive modeling details missing Expanded description, emphasized AUC < 0.65 Methods, Results, Discussion

Data availability Revised statement, summary data as SI Data Availability

Figures & Tables Renumbered, prepared for PACE All

Discussion Expanded for methodological points Discussion

Language Final proofread Whole text

Prof. Paweł Macek, PhD, MSc

Corresponding Author

Collegium Medicum, Jan Kochanowski University of Kielce, Poland

Scientific Research, Epidemiology and R&D Centre,

Holycross Cancer Centre, Kielce, Poland

pawel.macek@onkol.kielce.pl

---

## [Decision Letter · Decision Letter 1]

12 Nov 2025

Dear Dr. Macek,

Thank you for submitting your manuscript to PLOS ONE. After careful consideration, we feel that it has merit but does not fully meet PLOS ONE’s publication criteria as it currently stands. Therefore, we invite you to submit a revised version of the manuscript that addresses the points raised during the review process.

We look forward to receiving your revised manuscript.

Kind regards,

Stanisław Jacek Wroński, M.D., Ph.D, FEBU

Academic Editor

PLOS ONE

Journal Requirements:

Additional Editor Comments:

Dear Authors,

please find some remarks send by the reviewer. Taking into account special merits of this paper I send it back to you for minor revisions in the hope that, upon their appropriate incorporation, the manuscript will receive final approval.

with compliments

Stanisław Wroński

Academic Editor

Reviewers' comments:

Reviewer's Responses to Questions

**Comments to the Author**

Reviewer #2: All comments have been addressed

2. Is the manuscript technically sound, and do the data support the conclusions?

Reviewer #2: Yes

3. Has the statistical analysis been performed appropriately and rigorously?

Reviewer #2: I Don't Know

4. Have the authors made all data underlying the findings in their manuscript fully available?

Reviewer #2: Yes

5. Is the manuscript presented in an intelligible fashion and written in standard English?

Reviewer #2: Yes

Reviewer #2: Dear editor,

Thanks for choosing me as a reviewer;

While the authors have responded to the previous feedback, I have several additional questions and concerns for clarification, which I hope will strengthen the final text.

Abstract:

In the method section, it is not clear what the clustering is based on. Phenotype? Incontinence or...? Please explain fully.

In the results section, which session is meant by improvement in session 3? Is it the last session of the third part of the treatment? Please explain clearly.

Main text:

Include the code of ethics in the article

Line 98: What was the purpose of ultrasound imaging? This test is not mentioned in other parts of the text.

Were there surface electrodes on the rectus abdominis and gluteal to check for lack of help from other muscles? How many channels did the EMG device used have? And what is the name of the device?

In the exercises section, how did the patient maintain a 30-second contraction in the first treatment session!!

Were the exercises in the second and third sessions similar to sessions one? How many times were the treatment sessions per week? Explain more fully about the treatment sessions. How many sessions were the exercises in phases two and three and how many times a week?

Some abbreviations are not included in table one. For example, PSA - check the other tables to be sure.

The full name of the UPS is also not included in the text. Please check.

Thank you for inviting me to review this paper.

Abstract:

In the Methods section, it is unclear what the clustering was based on. Was it on phenotype, incontinence, or other factors? Please provide a full explanation.

In the Results section, which session is referred to by "improvement in third session "? Is this the last session of the third phase of treatment? Please clarify.

Main Text:

The ethics approval code should be included in the manuscript.

Line 97: What was the purpose of the ultrasound imaging? This assessment is not mentioned elsewhere in the text.

Were surface electrodes placed on the rectus abdominis and gluteal muscles to check for the absence of accessory muscle contraction? How many channels did the EMG device have, and what was the device model name?

In the exercises section, how were patients able to maintain a 30-second contraction during the first treatment session?

Were the exercises in the second and third phases similar to those in the first phase?How many times per week were the treatment sessions conducted? Please provide a more complete description of the treatment protocol, including how many sessions were in phases two and three and their weekly frequency.

Tables:

Some abbreviations in Table 1 are not defined (e.g., PSA). Please verify that all abbreviations are defined in all tables.

The full name of the "ISUP" is also not provided in the text. Please check and define all abbreviations upon first use.

**Do you want your identity to be public for this peer review?** For information about this choice, including consent withdrawal, please see our Privacy Policy

Reviewer #2: **Yes: ** seyedeh Saeideh Babazadeh-Zavieh

---

## [Author Response · Author response to Decision Letter 2]

18 Nov 2025

Response to Reviewer #2

Dear Reviewer,

We thank you very much for your thoughtful and constructive feedback. We appreciate the opportunity to further improve our manuscript. Below, we provide detailed responses to all comments, accompanied by the corresponding revisions to the text. All changes requested have been incorporated into the manuscript using Track Changes.

Abstract

Comment 1:

“In the method section, it is not clear what the clustering is based on. Phenotype? Incontinence or...? Please explain fully.”

Response:

Thank you for this important observation. We have revised the Methods section of the abstract to clarify the basis of the clustering. The sentence now reads:

“K-means clustering was applied to 11 baseline clinical variables, including urinary incontinence severity, pelvic floor function measures, and oncological risk characteristics, to identify distinct patient phenotypes.”

(See Abstract, Methods)

Comment 2:

“In the results section, which session is meant by improvement in session 3? Is it the last session of the third part of the treatment? Please explain clearly.”

Response:

Thank you for pointing this out. The text has been revised for clarity. We now specify that the third assessment refers to the rehabilitation evaluation conducted after the completion of phase III of the program. The revised sentence reads:

“By the third rehabilitation assessment (conducted after completing phase III of the rehabilitation program), continence rates increased to 88.4%, 77.5%, and 60.0% across the three clusters.”

(See Abstract, Results)

Main Text

Comment 3:

“Include the code of ethics in the article.”

Response:

We agree and have expanded the ethics statement to include reference to the Declaration of Helsinki. The revised sentence reads:

“The study was conducted in accordance with the Declaration of Helsinki and approved by the Bioethics Committee of Collegium Medicum, Jan Kochanowski University, Kielce, Poland (approval no. 34/2018). Written informed consent was obtained from all participants.”

(See Methods)

Comment 4:

“Line 97: What was the purpose of ultrasound imaging? This assessment is not mentioned elsewhere in the text.”

Response:

Thank you for highlighting this. We have now specified the purpose of ultrasound imaging in the Methods section. The revised sentence reads:

“Ultrasound sonofeedback was used during all sessions to provide real-time visualization of pelvic floor muscle contractions, enabling patients to learn correct activation and relaxation and to coordinate this activity with breathing.”

(See Methods, Rehabilitation protocol)

Comment 5:

“Were surface electrodes placed on the rectus abdominis and gluteal muscles to check for the absence of accessory muscle contraction? How many channels did the EMG device have, and what was the device model name?”

Response:

These details have now been added. Surface electrodes were placed on the rectus abdominis and right gluteal muscle to prevent accessory muscle activation. We also specified the EMG system used. The text now reads:

“Surface electrodes were placed over the rectus abdominis and right gluteal muscle to monitor and prevent the activation of accessory muscles during pelvic floor contractions. The electromyography system used in this study was a Noraxon Ultium® EMG device equipped with 16 channels, of which four were activated for the purposes of this protocol.”

(See Methods, Rehabilitation protocol)

Comment 6:

“In the exercises section, how were patients able to maintain a 30-second contraction during the first treatment session?”

Response:

We appreciate this question. Patients attempted a 30-second sustained contraction according to the standardized Glazer protocol implemented via Noraxon software. Not all were able to sustain the contraction for the full duration, as this task was used for functional assessment rather than as a therapeutic objective. The sentence has been added:

“While not all patients were able to maintain a full 30-second contraction during the initial session, this step served as a standardized functional assessment rather than a clinical outcome measure in this study.”

(See Methods, Detailed first session procedure)

Comment 7:

“Were the exercises in the second and third phases similar to those in the first phase? How many times per week were the treatment sessions conducted? Please provide a more complete description of the treatment protocol, including how many sessions were in phases two and three and their weekly frequency.”

Response:

Thank you for pointing this out. The text has been expanded to describe the exercise protocol in phases II and III. The revised sentence reads:

“In phases II and III of the rehabilitation program, patients attended five supervised physiotherapy sessions per week. In cases requiring adjunct neuromuscular electrostimulation, a total of 10 sessions were conducted over two weeks, also at a frequency of five sessions per week. The exercise protocol in these phases was similar to that used in phase I and included pelvic floor muscle activation and relaxation in lying, sitting, and standing positions, synchronizing contractions with breathing, and incorporating functional movement patterns.”

(See Methods, Rehabilitation protocol)

Tables

Comment 8:

“Some abbreviations in Table 1 are not defined (e.g., PSA). Please verify that all abbreviations are defined in all tables.”

Response:

Thank you for your careful review. We have updated the Abbreviations section under Table 1 to include the definition of PSA as “prostate-specific antigen.”

(See Table 1, Abbreviations)

Comment 9:

“The full name of the ‘ISUP’ is also not provided in the text. Please check and define all abbreviations upon first use.”

Response:

This has been corrected. The full name “International Society of Urological Pathology (ISUP)” is now provided upon first use in the text and included in the Abbreviations list under Table 1.

(See Methods and Table 1, Abbreviations)

We hope that these revisions adequately address all concerns. We are grateful for your valuable feedback, which has helped strengthen the clarity and rigor of our manuscript.

Sincerely,

Prof. Paweł Macek, PhD, MSc

Corresponding Author

Collegium Medicum, Jan Kochanowski University of Kielce, Poland

Scientific Research, Epidemiology and R&D Centre,

Holycross Cancer Centre, Kielce, Poland

pawel.macek@onkol.kielce.pl

on behalf of all authors

---

## [Decision Letter · Decision Letter 2]

30 Nov 2025

Phenotypic Stratification Predicts the Pace, but Not the Outcome, of Continence Recovery after Radical Prostatectomy

PONE-D-25-33254R2

Dear Dr. Paweł Macek

We’re pleased to inform you that your manuscript has been judged scientifically suitable for publication and will be formally accepted for publication once it meets all outstanding technical requirements.

Kind regards,

Stanisław Jacek Wroński, M.D., Ph.D, FEBU

Academic Editor

PLOS ONE

Dear Authors,

Considering that it took a long time to find a reviewer for this article I suggest that the article ultimately meets the conditions for publication in PLOS ONE.

Personally, I believe that the authors' approach to the issue of urinary incontinence after radical prostatectomy fully deserves attention and presentation in the journal.

With compliments

Stanisław Wroński

PLOS ONE academic editor

Reviewers' comments:

Reviewer's Responses to Questions

**Comments to the Author**

Reviewer #2: All comments have been addressed

2. Is the manuscript technically sound, and do the data support the conclusions?

Reviewer #2: Yes

3. Has the statistical analysis been performed appropriately and rigorously?

Reviewer #2: I Don't Know

4. Have the authors made all data underlying the findings in their manuscript fully available?

Reviewer #2: Yes

5. Is the manuscript presented in an intelligible fashion and written in standard English?

Reviewer #2: Yes

Reviewer #2: The authors have responded to the review and made corrections to the text. The manuscript appears to be suitable for acceptance.

**Do you want your identity to be public for this peer review?** For information about this choice, including consent withdrawal, please see our Privacy Policy

Reviewer #2: No

---

## [Editor Report · Acceptance letter]

PONE-D-25-33254R2

PLOS One

Dear Dr. Macek,

I'm pleased to inform you that your manuscript has been deemed suitable for publication in PLOS One. Congratulations! Your manuscript is now being handed over to our production team.

Kind regards,

on behalf of

Dr. Stanisław Jacek Wroński

Academic Editor

PLOS One